# Cerebellar functional connectivity alteration in individuals with lower limb amputation

Daniel Arreguín[1,2¤b], Daniela Trejo-Méndez[1,2¤a], Sharon Pedroza-Ramírez[1,2¤a], Erick Pasaye[1¤a], Livia Sánchez-Carrasco[3], Eduardo Vázquez-Vela[3], Raúl G. Paredes[1,4*¤a¤b]

1 Instituto de Neurobiología, Universidad Nacional Autónoma de México (UNAM) Campus Juriquilla, Querétaro, México, 2 Escuela Nacional de Estudios Superiores, Unidad León, UNAM, Guanajuato, México, 3 Crimal Hacienda Sta. Fe 110, El Jacal, 76180 Querétaro, México, 4 Escuela Nacional de Estudios Superiores, Unidad Juriquilla, UNAM, Querétaro, México

¤a Current Address:, Instituto de Neurobiología, Universidad Nacional Autónoma de México (UNAM) Campus Juriquilla, Querétaro, México.
¤b Current Address: Escuela Nacional de Estudios Superiores, Unidad Juriquilla, UNAM, Querétaro, México.
* rparedes@unam.mx

## Abstract

Limb amputation triggers a reorganization of brain structure and function. Previous research using neuroimaging techniques, such as resting state functional magnetic resonance imaging, indicates reduced functional connectivity in the sensorimotor network in amputees compared to non-amputees. Yet, data on lower limb amputees remains limited. We investigated functional connectivity differences within the sensorimotor network (S1M1) using an analysis of 10 regions of interest in a group of 26 lower limb amputees compared to non-amputees. The statistical analysis revealed a decreased connectivity network component involving cerebellar regions within the S1M1 network, specifically between the contralateral Cerebellum and primary sensorimotor cortical areas ipsilateral to the amputated limb. These findings underscore the complexity of cerebral adaptations post-amputation, highlighting a significant decrease in functional connectivity at network level. The research emphasizes the importance of investigating changes in functional connectivity in this population to understand the neuroadaptive processes resulting from an amputation.

## Introduction

Limb amputations are traumatic events that extend beyond the loss of physical abilities, impacting both brain function and the quality of life of affected individuals. The brain's ability to adapt to such anatomical alterations has become a subject of increasing interest in neuroscientific research. Numerous studies have demonstrated that amputations induce plastic changes in the structure and function of the brain,

**Data availability statement:** ****A/PROS AT ACCEPT: Please follow up with the authors for external data repository access information at Accept.***** We are in the process of making the data available on GitHub.com or openneuro.org.

**Funding:** The Laboratorio Nacional de Imagenología por Resonancia Magnética (LANIREM) for providing access to the magnetic resonance scanner. Our appreciation also goes to the Unidad de Órtesis y Prótesis de la Escuela Nacional de Estudios Superiores (ENES) Juriquilla and other institutions facilitating participant access. Daniel Arreguín, a PhD student in the Psychology program at the Universidad Nacional Autónoma de México (UNAM), received scholarship support (number 1084993) from the National Council for Humanities, Science and Technology (CONAHCYT). Research supported by PAPIIT UNAM IN214524 and CONCYTEQ/CACTI/094/2024 The funders had no role in study design, data collection and analysis, decision to publish, or preparation of the manuscript.

**Competing interests:** The authors have declared that no competing interests exist.

triggering remarkable adaptations that can be explored through advanced neuroimaging techniques.

Cerebral plastic changes research in amputated individuals has focused on various magnetic resonance imaging (MRI) techniques. Diffusion tensor imaging, for instance, has revealed decreases in diffusion parameters in axonal tracts of the corpus callosum, connecting supplementary motor areas bilaterally in individuals with lower limb amputations [1]. Volumetric MRI has been employed to assess changes in cortical region volume, indicating decreases in visual, sensory, and motor areas post-amputation [2–3]. Compared to non-amputated controls, cerebellar volume reduction has also been noted in non-prosthesis users [4]. Cortical thickness measurements have uncovered correlations, such as the volume of visual regions correlating with prosthetic use in individuals experiencing phantom limb pain [5]. Functional neuroimaging studies have revealed altered cortical activation patterns in individuals with amputations. These findings suggest that the brain undergoes significant functional reorganization following limb loss, particularly within the cortical maps representing the body [6–8]. The authors propose that the loss of afferent information, time elapsed since amputation, prosthesis use, and even amputation techniques contribute to cerebral modifications in the amputated population, emphasizing the need for more studies under different conditions to identify underlying mechanisms and characteristic plastic patterns [1–9].

The study of resting state functional connectivity of different networks has gained significance in the amputees population, particularly in studying changes in the sensorimotor network (S1M1). This network, crucial for integrating sensory and motor information, has been the subject of numerous investigations aimed at identifying adaptations in individuals with amputations under various conditions. Functional brain connectivity measurement, conducted through functional magnetic resonance imaging (fMRI) using blood oxygen level-dependent (BOLD) signal temporal series, enables exploration of correlations between different regions during resting conditions [10,11]. For example, Chicos et al. [9] reported differences in resting-state functional connectivity between subjects with transtibial amputation who underwent a novel amputation procedure called the agonist-antagonist myoneural interface compared to subjects who underwent traditional amputation methods where subjects who underwent this new procedure exhibited reduced connectivity with the salience network. The authors suggested that this procedure facilitates the preservation of proprioceptive efficiency in individuals after amputation and that the reduced cognitive load in this network could contribute to improvements in prosthetic use.

Zhang et al. [12] reported a significant decrease in the functional connectivity of the S1M1 network, encompassing connections between cortical and subcortical structures bilaterally in unilateral lower limb amputees, users, and non-users of prosthesis. They also identified a correlation between the connectivity of these network connections and the time elapsed since amputation, highlighting the temporal dynamics of these changes. Additionally, Makin et al. [13] revealed a network-level reorganization, observing a decrease in connectivity between the S1M1 network and the cortical representation of the amputated limb, as well as an increase in connectivity

between this representation and the default mode network. This reorganization was more pronounced in individuals with a longer time since amputation, suggesting continuous adaptation over time. Furthermore, the study by Zheng et al. [14] found a negative correlation between scores obtained in the SF-36 quality of life questionnaire and increased functional connectivity of the supplementary motor area with subcortical structures in individuals reporting painful perceptions of phantom sensations. These findings suggest that phantom pain might be influenced by changes in the connectivity of structures and regions comprising the sensorimotor network, providing a unique perspective on the functional implications of such adaptations on the quality of life of amputated individuals.

These background findings underscore that understanding both the structural and functional modifications of the brain following amputation is essential to comprehend adaptive brain mechanisms and how these changes influence the sensory and motor expressions of affected individuals. In this context, the importance of research exploring the functional connectivity of brain networks through resting-state magnetic resonance imaging is emphasized, providing valuable information on the neurobiological mechanisms linked to sensory and motor adjustments, whether in the short or long term, in individuals with limb amputations.

This study aimed to identify differences in the functional connectivity of the S1M1 network between individuals with unilateral lower limb amputation and non-amputated individuals, using resting-state functional magnetic resonance imaging. We hypothesized that there would be a network component with decreased functional connectivity within this network in participants with amputation compared to participants without amputation.

## Materials and methods

### Participants

A cohort of twenty-six individuals with unilateral lower limb amputation occurring above the ankle (average age: 50.23 ± 12.14 years; 19 males and 7 females; 19 with left-sided amputation; 12 prosthetic users; see Table 1), was selected from a database comprising a total of 65 participants recruited from rehabilitation centers and hospitals in the city of Querétaro, México; spanning the years 2021–2024. Out of the twenty-six amputee group (AMP), eighteen participants had a transfemoral amputation level, seven had a transtibial level amputation, and one had a hemipelvectomy. The etiology of amputation was vascular complications resulting from diabetes mellitus in twenty-one participants, traumatic causes in four, and a single case due to cancer. All participants had an elapsed time since amputation equal to or less than 36 months. Exclusion criteria included having more than one amputation, neurological or psychiatric diseases diagnosed, a history of drug consumption, and possessing devices or implants incompatible with magnetic resonance studies. The non-amputated control group (HC) was composed of twenty-six individuals without amputation, matched in age. Handedness was assessed for all participants, and it was confirmed that every member of both groups was right-handed using the Edinburgh Handedness Inventory [15]. These participants were recruited from the local community through direct contact and social media advertising. All participants were briefed on the study's purpose, signed informed consent

**Table 1. S1M1network ROI MNI Coordinates.**

| ROI | Ipsilateral MNI Coordinates | Contralateral MNI Coordinates |
|---|---|---|
| M1 | −8, −26, 62 | 2, −32, 62 |
| S1 | −20, −42, 62 | 16, −36, 72 |
| SMA | −2, −14, 64 | 2, −6, 52 |
| CerIV-V | −18, −38, −28 | 18, −38, −26 |
| CerVI | −30, −74, −21 | 27, −60, −21 |

MNI Coordinates are expressed in mm. **M1** refers to the Primary Motor Area, **S1** to the Primary Somatosensory Area, **SMA** to the Supplementary Motor Area and **Cer** to the Cerebellum.

forms, and underwent a clinical history questionnaire related to amputation. The questionnaire included variables such as amputation etiology, amputation level, amputation laterality, time since amputation, and prosthesis usage. The protocol (083.H-RM) was approved by the Ethics Committee of the Institute of Neurobiology at UNAM, which follows the guidelines of the NIH.

## Magnetic resonance imaging acquisition

Structural and resting-state functional images were acquired using a General Electric 3.0 T Discovery MR750 scanner with a 32-channel head coil. Resting-state functional images were obtained through an axial echo-planar imaging (EPI-GR) sequence with the following parameters: 300 volumes (600 sec); repetition time (TR) = 2000 ms; echo time (TE) = 40 ms; flip angle = 90°; matrix = 64 x 64; slice thickness = 4 mm; field of view = 256 mm$^2$; 35 slices with a voxel size = 4 x 4 x 4 mm. Participants were asked to fixate on a cross, avoid head movement, and abstain from focusing on specific thoughts during image acquisition to minimize experimental variability [16]. T1-weighted structural images were acquired using an axial 3D BRAVO sequence. The following parameters where used: TR = 6.9 ms, TE = 2.9 ms, flip angle = 12°, matrix = 256 x 256, FOV = 256 mm$^2$, slice thickness = 1 mm, 312 slices with a voxel size = 1 x 1 x 1 mm. Additionally, reversed phase-encoded images AP-PA were obtained for field maps calculation [17,18] using a diffusion sensitive sequence with the following parameters: TR = 4000 ms, TE = 78 ms, flip angle = 90°, slice thickness = 2 mm, matrix = 128 x 128, FOV = 256 mm$^2$ and voxel size = 2 x 2 x 2 mm.

## Image preprocessing

Before preprocessing, phase-reversed structural and functional images of participants with a right-sided amputation underwent hemisphere inversion across the sagittal midline. This step aimed to standardize the analysis concerning the contralateral side of the amputation in all participants. Also, field maps were calculated with the phase reversed images. These steps were executed using the FSL software (http://www.fmrib.ox.ac.uk/fsl/, RRID: SCR_002823) tools *fslswapdim* and *topup*, respectively [19,20]. For image preprocessing, we used the CONN toolbox v22.a (https://web.conn-toolbox.org/, RRID: SCR_009550) for Matlab vR2023a [21,22]. This toolbox relies on the tools package of the SPM software v12.7771 [23] (RRID: SCR_007037).

Functional and structural images and field maps were imported into CONN and underwent the default preprocessing pipeline for indirect normalization to the Montreal Neurological Institute (MNI) standard space [24] which is included in the software. This pipeline generated slice-timing correction (STC), voxel displacement maps for motion and susceptibility-induced distortion corrections through field maps, outlier detection, indirect segmentation, normalization to the MNI space, and subsequent smoothing. Functional data alignment utilized the SPM realignment and unwarping procedure [25], integrating field maps for magnetic susceptibility distortion correction. Co-registration involved a least squares approach and rigid body transformation with b-spline interpolation for motion and geometric distortion correction using the structural data. The STC step from SPM [26] addressed temporal misalignment between slices acquired in interleaved ascending order. Outlier scans were identified using the Artifact Detection Tool (ART) based on frame displacement with a subject motion correction FWD = 2.5 mm and percentage of changes in global BOLD signal with a Z-value threshold ≥ 3, no participants were excluded from the analysis. Functional and anatomical data were then registered and normalized to the standard MNI space, segmented into gray matter, white matter, and cerebrospinal fluid (CSF) tissue classes, and resampled to isotropic 2 mm voxels [27,28]. Lastly, functional data were smoothed using spatial convolution with a 4 mm Gaussian kernel. Noise reduction in functional data utilized a conventional denoising procedure based on anatomical component correction (aCompCor) [29]. Confounding effects were regressed, including white matter and CSF temporal series, motion parameters and their derivatives, outlier scans, session effects and their derivatives, and linear trends. A band-pass filter between 0.008 and 0.09 Hz was then applied to isolate the neuronal activity signal of interest [30].

## ROI selection and functional connectivity analysis

To evaluate the functional connectivity of the S1M1 network, a mask consisting of 10 spherical regions of interest (ROIs), each with a radius of 5 mm, was generated. This mask was created using the FSLeyes image viewer editing tools within the MNI space. The selection of coordinates for each ROI was based on prior studies reporting peak cortical activation associated with lower limb movement and sensation [31,32] using the Neurosynth search engine [33] with the keyword "lower limb," coordinates were cross-referenced with the Harvard-Oxford cortical and AAL subcortical atlases included in CONN.

ROIs were delineated ipsilateral and contralateral to the amputation using MNI coordinates (Table 1 and Fig 1). The average time series for each ROI were extracted, and ROI-ROI connectivity matrices (10 x 10) were generated. Functional connectivity strength was represented by r to Z Fisher's-transformed scores obtained from a bivariate correlation coefficients analysis with a general linear model, estimated separately for each pair of ROIs for the 45 possible connections in each group. These transformed values were used for network-level and connection-level connectivity difference analysis between groups, employing the Network-Based Statistics (NBS) method [34] integrated into the second-level analysis of CONN.

The Network-Based Statistic (NBS) is a non-parametric method designed to identify differences in brain connectivity at the level of interconnected subnetworks rather than isolated connections [35]. This approach is particularly well-suited for studies with small or unbalanced samples, as it leverages permutation testing to control the family-wise error rate (FWE) while maintaining statistical power. NBS uses a *t*-statistic derived from a general linear model (GLM) to test for between-group differences in the strength of functional connectivity between regions. Connections that exceed a predefined threshold are grouped into connected components (i.e., subnetworks), and their significance is evaluated through 1,000 permutations of group labels to generate a null distribution of network sizes (The suprathreshold connections constituting the identified subnetwork) or intensities (mass). Unlike mass univariate tests that evaluate each connection independently, NBS detects topologically connected subnetworks of connections (clústers) that collectively show significant effects. In this study, NBS was applied to assess group differences in resting-state functional connectivity. The significance level for connectivity analysis was set at *p*-FDR < 0.05 at the connections level and *p*-FWE < 0.05 for network mass and intensity measures.

## Clinical and demographic data analysis

Clinical and demographic data were analyzed using R Studio software v.2023.6.1.524 [36]. Normal distribution and variance homogeneity were analyzed using Shapiro-Wilk and Levene tests, respectively. The Mann-Whitney U test was used

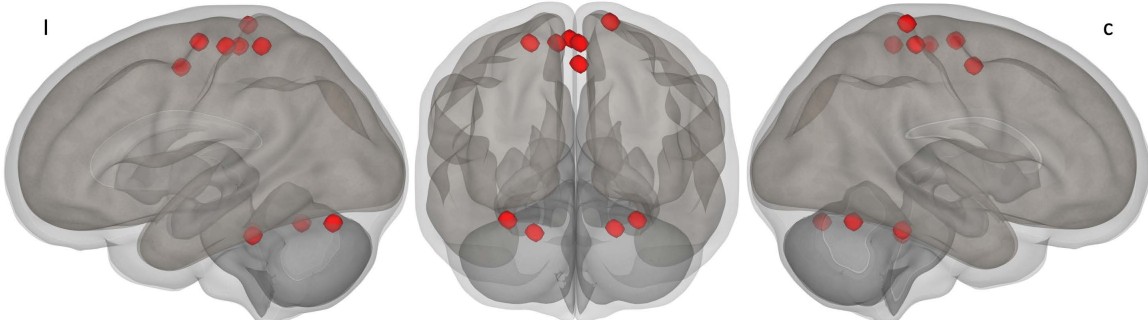

**Fig 1. 3D visualization of S1M1 Network ROIs.** I, Ipsilateral hemisphere to the amputation; C, Contralateral hemisphere to the amputation. The network consists bilaterally of the following regions: **M1**, primary motor area; **S1**, primary sensory area; **SMA**, supplementary motor area; CerIV-V, cerebellum IV-V; CerVI, cerebellum VI.

to compare age and years of education between both groups. The difference in proportions regarding sex was evaluated using the chi-square test. The significance level for all tests was set at $p < 0.05$.

## Results

### Clinical and demographic characteristics

Table 2 summarizes the demographic and clinical characteristics of both the lower limb amputees group (LLA) and the healthy control group (HC). While no significant differences were observed in age, the groups differed significantly in years of education and sex proportions ($p < 0.05$)

### ROI-to-ROI functional connectivity analysis

To assess group differences in functional connectivity, we applied the NBS method, as implemented in the CONN Connectivity Toolbox. This analysis focused on connectivity patterns among 10 regions of interest (ROIs), each with a 5 mm radius, comparing the LLA group with the HC group. Our investigation identified a significant subnetwork (mass = 64.55, size = 4, p-FWE < 0.05), which indicated altered connectivity between the groups. This subnetwork was specifically composed of two anticorrelated connections: M1.i-CerVI.c ($T_{(50)}$ = −4.11, p-FDR < 0.05) and S1.i-CerVI.c ($T_{(50)}$ = −3.92, p-FDR < 0.05). As detailed in Table 3 and Fig 2, these connections exhibited a substantial decrease in connectivity within

**Table 2. Clinical and demographic characteristics of participants.**

| Feature | LLA (n = 26) | HC (n = 26) | Statistic | p-value |
|---|---|---|---|---|
| Age (years) | 52 (28/ 68) | 46.5 (21/ 68) | 277 | 0.268[a] |
| Education (years) | 11.5 (3/ 19) | 15 (6/ 26) | 186 | 0.004[a] |
| Sex (M/ F) | 19/ 7 | 12/ 14 | 1.98 | 0.048[b] |
| Hand dominance (R/ L) | 26/ 0 | 26/ 0 | – | – |
| Time since amputation (months) | 13 (1/ 36) | – | – | – |
| Amputation level (TT/ TF/ HP) | 7/ 18/ 1 | – | – | – |
| Amputation side (R/ L) | 7/ 19 | – | – | – |
| Amputation etiology (DM/ TR/ CA) | 21/ 4/ 1 | – | – | – |
| Prosthesis users (Yes/ No) | 12/ 14 | – | – | – |

*Data are expressed as medians (min/max). Statistics and p-values were calculated using: [a]Mann-Whitney U test and [b]Chi-square test. **LLA** refers to the lower limb amputation group, **HC** to the healthy control group, **M** to male participants, **F** to female participants; **TT** to transtibial level, **TF** to transfemoral level, **HP** to hemipelvectomy level, **R** to right hand dominance, **L** to left hand dominance, **DM** to diabetes mellitus type 2, **TR** to trauma and **CA** to cancer.*

**Table 3. Results of functional connectivity differences between groups in the S1M1 network.**

| Network | Statistic | p-value | p-FDR | p-FWE | |
|---|---|---|---|---|---|
| S1M1. | **Mass** = 64.55 | 0.0092 | 0.0092 | 0.0090 | |
| | **Size** = 4 | 0.0180 | 0.0180 | 0.0190 | |
| Connections | LLA (n = 26) | HC (n = 26) | Statistic | p-value | p-FDR |
| M1.i – CerVI.c | −0.1239 ± 0.1334 | −0.0261 ± 0.1295 | $T_{(50)}$ = −4.11 | 0.0001 | 0.0061 |
| S1.i – CerVI.c | −0.1465 ± 0.1789 | −0.0233 ± 0.1301 | $T_{(50)}$ = −3.92 | 0.0002 | 0.0061 |

*Differences at both the network level and individual connection level were calculated using the **NBS** method. Connectivity values for individual connections are presented as the mean ± standard deviation (SD) of **Fisher's r-to-z transformed scores**. **LLA** refers to the Lower Limb Amputee group, **HC** to the Healthy Controls group, **M1** to the Primary Motor Area, **S1** to the Primary Somatosensory Area, and **Cer** to the Cerebellum. Laterality is denoted by **i** for ipsilateral to the amputation and **c** for contralateral to the amputation. Statistical corrections included False Discovery Rate (FDR) and Family-Wise Error (FEW) to address multiple comparisons.*

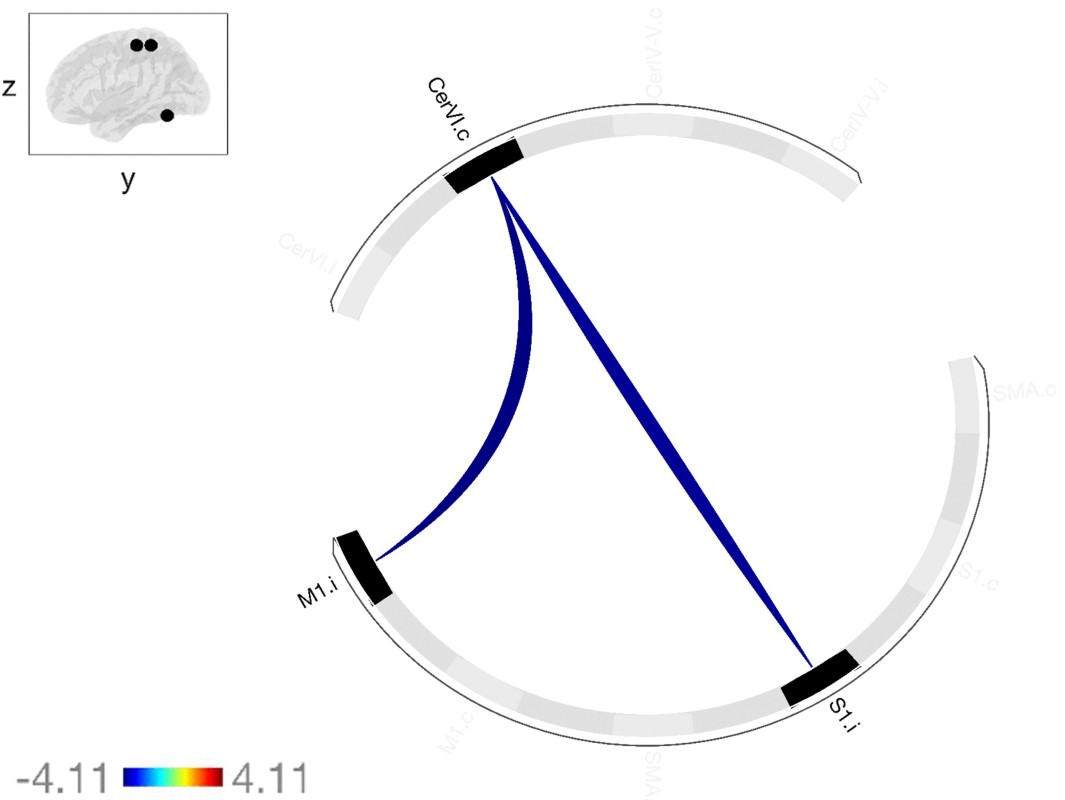

**Fig 2. Significant network component showing decreased functional connectivity in Lower Limb Amputees Compared to Healthy Controls (LLA > HC).** This connectogram displays a significant network component identified by NBS showing decreased functional connectivity in the LLA group compared to HC. The component consists of anticorrelated connections (blue edges) between the CerVI.c-S1.i ($T_{(50)}$= −4.11, p-FDR < 0.05), and between CerVI.c-M1.i ($T_{(50)}$= −3.92, p-FDR < 0.05). Nodes represent brain regions. The network component achieved family-wise error corrected significance (p-FWE < 0.05). **Cer** refers to Cerebellum, **S1** to the primary sensory area. **M1** to the primary motor area, **c** to the contralateral to the amputation, **i** to the ipsilateral to the amputation.

the LLA group. A 3D visualization of this subnetwork is presented in Fig 3. For more details about functional connectivity characterization within groups, please refer to supporting information S1 and S2 Tables.

## Discussion

In this study on brain functional connectivity in individuals with lower limb amputation, using the NBS method, a significant network component of altered connectivity within the S1M1 network was identified. Our analysis revealed two anticorrelated connections showing a significant reduction in their connectivity patterns in the LLA group compared to the HC group (p-FDR < 0.05). Specifically, these connections were S1.i–CerVI.c and M1.i–CerVI.c, both converging on the CerVI.c node. The convergence of these two connections on a central node forming an interconnected subnetwork that reaches significance at the network component level (*p*-FWE < 0.05) highlights a focal neurofunctional adaptation in this sample.

This study observed a decrease in functional connectivity between cerebellar and cortical regions involved in sensorimotor processing, which aligns with previous findings. For example, Zhang et al. [12] also reported reduced cerebellar connectivity in similar connections within the S1M1 network. Notably, the finding emerges as a subnetwork composed of anticorrelations between ROIs, indicating reciprocal neuronal activity, where an increase in activity in one region corresponds to a decrease in activity in another. Importantly, this temporally inverse pattern does not imply a lack of interaction,

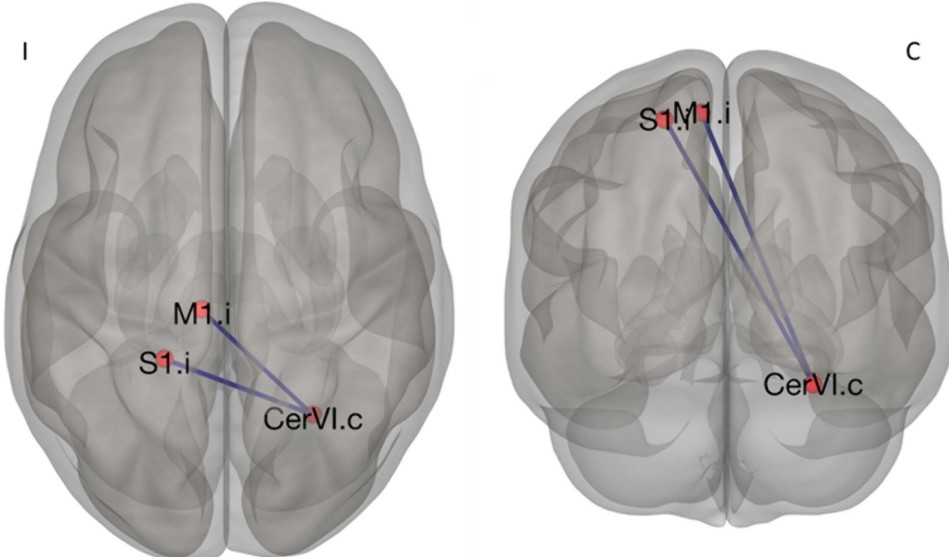

**Fig 3. 3D visualization of the decreased functional connectivity component network within the S1M1 network in the LLA group compared to the HC group.** Blue lines represent connections that showed decreased connectivity (p-FDR < 0.05). **M1** refers to the Primary Motor Area, **S1** to the Primary Sensory Area, **SMA** to the Supplementary Motor Area, **Cer** to the Cerebellum, **I** to Ipsilateral to amputation and **C** to Contralateral to amputation.

but rather an active functional modulation [37]. Although the exact interpretation of resting-state fMRI anticorrelations remains under debate, some studies suggest they may be related to neuronal inhibitory mechanisms that reflect the modular coordination in neural resting-state activity [38,39].

Generally, limb loss requires a process of motor learning to acquire or retrain functional skills and movement patterns during rehabilitation [40]. Given the Cerebellum's essential role in motor learning, planning, coordination, execution, and error correction of movement patterns [41,42], it is possible that the altered connectivity patterns observed in this study may affect motor and sensory integration in the LLA group. In this regard, other studies [2,3,5,12,15] have shown that changes in cortical thickness, white matter integrity, and functional connectivity are related to the type of prosthesis used. These studies propose that motor learning involved in prosthetic use is one of the key factors driving brain plasticity in this population.

Furthermore, cerebellar motor control is ipsilateral, while cerebello-cortical pathways decussate via the thalamus before reaching the cortex. Thus, it is important to note that the altered connections identified in our study involve regions theoretically preserved in the LLA group, as they contribute to the motor control of the intact limb. Therefore, the identified subnetwork composed by S1.i–CerVI.c and M1.i–CerVI.c may suggest that the connectivity alteration reflects possible motor compensations in the LLA group. This could indicate a complex functional reorganization aimed at optimizing the control of the intact limb, which may become increasingly critical for stability, balance, and gait post-amputation [43,44]. However, confirming the functional implications of this adaptation would require targeted behavioral and clinical assessments.

It is essential to acknowledge that the cross-sectional design of this study is an inherent limitation, as it only captures group differences at a single time point. This constrains the ability to establish a direct causal relationship between amputation and the observed connectivity changes. To understand the trajectory of these neurofunctional adaptations and their association with the amputation experience, longitudinal studies incorporating pre and post-rehabilitation assessments, including clinical and behavioral evaluations, will be essential.

Sample size and heterogeneity also remain important considerations. While demographic and clinical variables were carefully evaluated using non-parametric statistics, variability in factors such as years of education, time since amputation, and prosthesis use may still influence results. These limitations highlight the need for future studies with larger sample sizes and more robust strategies to control for potential confounders. Despite these limitations, this study offers valuable insights into neurofunctional adaptations following lower limb amputation, particularly those involving motor control of the intact limb. It is important to acknowledge that the present analysis did not incorporate potential cortical reorganization before the evaluation of functional connectivity, a factor that should be considered in future projects. Integrating clinical and behavioral assessments will be essential to establish meaningful correlations between functional connectivity changes and improvements in motor and cognitive functions throughout rehabilitation. Furthermore, examining the influence of additional amputation-related variables, such as surgical technique, on brain plasticity and functional outcomes may further enhance our understanding of post-amputation neural reorganization and contribute to the development of novel therapeutic strategies that address the integration of both motor and higher-order cognitive processes.

## Supporting information

**S1 Table. S1M1 network functional connectivity characterization results for the HC group.** The values in the table represent **t-scores** for individual functional connections within the S1M1 network, reflecting their statistical significance. A higher absolute t-score indicates a stronger and more statistically robust connection. At the network level, two additional metrics are presented: **Mass** and **Size**. Mass represents the total magnitude of connectivity within the network. Size indicates the number of statistically significant connections, providing a measure of the network's extent. **M1** refers to the primary motor area, **S1** to the primary sensory area, **SMA** to the supplementary motor area, **Cer** to the Cerebellum, **i** to ipsilateral to amputation, **c** to contralateral to amputation, **S1M1** to sensorimotor network, **FDR** to false discovery rate correction, and **FWE** to family wise error correction.
(DOCX)

**S2 Table. S1M1 network functional connectivity characterization results for the AMP group.** The values in the table represent **t-scores** for individual functional connections within the S1M1 network, reflecting their statistical significance. A higher absolute t-score indicates a stronger and more statistically robust connection. At the network level, two additional metrics are presented: **Mass** and **Size**. Mass represents the total magnitude of connectivity within the network. Size indicates the number of statistically significant connections, providing a measure of the network's extent. **M1** refers to the primary motor area, **S1** to the primary sensory area, **SMA** to the supplementary motor area, **Cer** to the Cerebellum, **i** to ipsilateral to amputation, **c** to contralateral to amputation, **S1M1** to the sensorimotor network, **FDR** to false discovery rate correction, and **FWE** to family wise error correction.
(DOCX)

## Acknowledgments

To the Laboratorio Nacional de Imagenología por Resonancia Magnética (LANIREM) for providing access to the magnetic resonance scanner. Our appreciation also goes to the Unidad de Órtesis y Prótesis de la Escuela Nacional de Estudios Superiores (ENES) Juriquilla and other institutions facilitating participant access.

## Author contributions

**Conceptualization:** Daniel Arreguin, Sharon Pedroza-Ramirez, Daniela Trejo-Mendez, Eduardo Vazquez-Vela, Raúl G. Paredes.

**Formal analysis:** Daniel Arreguin, Sharon Pedroza-Ramirez, Daniela Trejo-Mendez, Erick Pasaye, Livia Sanchez-Carrasco.

**Funding acquisition:** Raúl G. Paredes.

**Investigation:** Daniel Arreguin, Sharon Pedroza-Ramirez, Daniela Trejo-Mendez, Raúl G. Paredes.

**Methodology:** Daniel Arreguin, Sharon Pedroza-Ramirez, Daniela Trejo-Mendez, Erick Pasaye, Livia Sanchez-Carrasco, Eduardo Vazquez-Vela, Raúl G. Paredes.

**Resources:** Raúl G. Paredes.

**Software:** Erick Pasaye.

**Supervision:** Livia Sanchez-Carrasco, Raúl G. Paredes.

**Validation:** Eduardo Vazquez-Vela.

**Writing – original draft:** Daniel Arreguin, Daniela Trejo-Mendez.

**Writing – review & editing:** Sharon Pedroza-Ramirez, Erick Pasaye, Livia Sanchez-Carrasco, Eduardo Vazquez-Vela, Raúl G. Paredes.

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
