## [Decision Letter · Decision Letter 0]

17 Mar 2025

Dear Dr. Paredes,

Thank you for submitting your manuscript to PLOS ONE. After careful consideration, we feel that it has merit but does not fully meet PLOS ONE’s publication criteria as it currently stands. Therefore, we invite you to submit a revised version of the manuscript that addresses the points raised during the review process.

We look forward to receiving your revised manuscript.

Kind regards,

Usman Ghafoor

Academic Editor

PLOS ONE

“Research supported by PAPIIT UNAM IN214524 and CONCYTEQ/CACTI/094/2024 to RP.”

“The Laboratorio Nacional de Imagenología por Resonancia Magnética (LANIREM) for providing access to the magnetic resonance scanner. Our appreciation also goes to the Unidad de Órtesis y Prótesis de la Escuela Nacional de Estudios Superiores (ENES) Juriquilla and other institutions facilitating participant access. Daniel Arreguín, a PhD student in the Psychology program at the Universidad Nacional Autónoma de México (UNAM), received scholarship support (number 1084993) from the National Council for Humanities, Science and Technology (CONAHCYT). Research supported by PAPIIT UNAM IN214524 and CONCYTEQ/CACTI/094/2024.”

“Research supported by PAPIIT UNAM IN214524 and CONCYTEQ/CACTI/094/2024 to RP.”

4. In the online submission form you indicate that your data is not available for proprietary reasons and have provided a contact point for accessing this data. Please note that your current contact point is a co-author on this manuscript. According to our Data Policy, the contact point must not be an author on the manuscript and must be an institutional contact, ideally not an individual. Please revise your data statement to a non-author institutional point of contact, such as a data access or ethics committee, and send this to us via return email. Please also include contact information for the third party organization, and please include the full citation of where the data can be found.

6. We note that there is identifying data in the Supporting Information file <file name>. Due to the inclusion of these potentially identifying data, we have removed this file from your file inventory. Prior to sharing human research participant data, authors should consult with an ethics committee to ensure data are shared in accordance with participant consent and all applicable local laws.

-Location data

Additional Editor Comments:

Based on reviewers comments, it is difficult to accept the mansucript in the current form.

The authors are encouraged to rebuttal and make major changes to the mansucript for re-evaluation.

Reviewers' comments:

Reviewer's Responses to Questions

**Comments to the Author**

1. Is the manuscript technically sound, and do the data support the conclusions?

Reviewer #1: Partly

Reviewer #2: Partly

2. Has the statistical analysis been performed appropriately and rigorously?

Reviewer #1: No

Reviewer #2: No

3. Have the authors made all data underlying the findings in their manuscript fully available?

Reviewer #1: No

Reviewer #2: Yes

4. Is the manuscript presented in an intelligible fashion and written in standard English?

Reviewer #1: Yes

Reviewer #2: No

Reviewer #1: “Cerebellar modifications of brain functional connectivity in individuals with lower limb amputation” by Arreguin, et al., reports on an investigation of functional connectivity differences between a group of 26 individuals with lower limb amputation, compared with 26 matched healthy control participants, using resting-state functional MRI. The reported finding of significant anticorrelation between a spherical region of interest designated as ipsilateral primary motor cortex and a spherical region of interest designated as contralateral cerebellar lobule VI may be of some interest.

However, it is not possible to recommend publication of the manuscript as submitted, for the following reasons:

1. The conclusions stated are not supported by the results presented, as most of the reported results fail to reach statistical significance.

2. The finding of significant anticorrelation does not appear to comport with the definition of functional connectivity, and thus requires proper interpretation.

3. The use of the “network-based statistic” (NBS) methodology to compute only correlations between pairs of regions, rather than larger networks, appears suboptimal.

4. The overall approach—the analytic methodology used, and the interpretation of results—neglects an ample literature indicating that amputation generally results in “massive functional reorganization in somatosensory cortex” (see below).

5. The submission is not in compliance with the PLOS ONE data sharing requirement.

Below we elaborate on these five points, and then list several other concerns.

MAJOR CONCERNS:

1. Most results fail to reach significance.

Table 3 lists "differences in ROI-to-ROI functional connectivity between [groups]" but only the first item survives correction for multiple comparisons, and is therefore statistically significant. The rest are not.

Thus, claims in the abstract that "statistical analysis revealed decreased connectivity in connections…” (note: “connections,” plural) and that "findings ... [highlight] a significant decrease in functional connectivity between the contralateral cerebellum and primary sensorimotor areas…” (note: “areas,” plural) are not supported by the results presented, because only one inter-regional connection was found to differ significantly between groups.

The use of ten regions of interest yields 45 total possible inter-regional connections, so using a p level threshold of 0.05, without correction for multiple comparisons, the analysis of pure noise would be expected to generate on average 0.05 x 45 = 2.25 apparent but not actually significant differences between groups. Of course, repeating this thought experiment with a fresh set of pure noise would yield a similar number of apparent but not truly significant differences between groups—but they would be overwhelmingly likely to be a different set of inter-regional correlations; such results would not be expected to replicate.

Table 3 of the current manuscript lists one correlation that is statistically significant (because it survives correction for multiple comparisons), plus five more, which are not statistically significant. That the authors report five such insignificant correlations, whilst the expected average number is 2.25, is of little interest. These same findings are graphically depicted in Figures 2 and 3; again, most of what is shown here is not statistically significant, and such insignificant findings cannot properly support any scientific conclusions.

Finally, the title of the manuscript is not supported by the results presented. Since only one “connection” is significantly different between the amputee group and the baseline group, the title’s references to “modifications”—plural—is not appropriate.

2. Anticorrelation isn’t connectivity.

The statistically significant finding is of an anticorrelation, but the negative sign—the finding of a negative correlation coefficient—is not mentioned or discussed. Functional connectivity is defined as synchronous neural activity in spatially remote locations, corresponding to correlation. However, the reported finding is an anticorrelation—a correlation coefficient with a negative sign. This does not comport with the definition of functional connectivity. Instead, it resembles the well-known temporal anticorrelation of activity in the default mode network with respect to activity in task-positive regions. The finding is of some interest, but its sign is important—the anticorrelation should be presented and discussed as such, not ignored and treated as if it were a positive correlation.

3. The network-based statistic is questionable when restricted to pairs of regions.

The network-based statistic is analogous to the (more familiar and well-established) cluster-based statistic. The idea of the cluster-based statistic is that rather than testing the null hypothesis at every voxel, we test the null hypothesis for every cluster of potentially active voxels. The advantage is more power; the disadvantage is that we don’t know just where in the cluster the activity is—we know there is activity in the cluster, but not precisely where in the cluster.

The network-based statistic is similar: Instead of testing the null hypothesis at every voxel, we test the null hypothesis for every network of voxels (or, more generally, nodes). This yields more power, but at the expense of not knowing just where within an “active” network the “activity” actually is. We can say the network is active (or significant), but not where within the network the activity or significance resides.

What is key in both cases is the use of clusters or networks of substantial size. It would be odd to use cluster-based statistics but limit analysis to clusters comprised of just two voxels. However, unless this reviewer is mistaken, the present manuscript appears to do this with the network-based statistic. It appears that only correlations between two regions are considered; these are networks comprised of only two nodes. This seems a suboptimal use of the network-based statistic.

4. “Remapping” has been ignored.

It is well-established that “Peripheral denervations of large extents of the sensory epithelium, as a result of limb amputation … can lead to massive functional reorganization in somatosensory cortex…” (from “Thalamic and Cortical Contributions to Neural Plasticity After Limb Amputation,” by Florence, Hackett, and Strata, J. Neurophysiology (2000) https://doi.org/10.1152/jn.2000.83.5.3154

Indeed, the introductory section of the present manuscript (p. 10) refers to “…post-amputation cortical functional reorganization…”

However, the analysis in the present manuscript ignores such functional reorganization, as it is limited to computing correlations between spatially-smoothed MR signal averaged over ten-cm spheres located in fixed positions in neuroanatomical atlas space. The “massive functional reorganization in somatosensory cortex” of the individuals with amputation is simply ignored. This is a suboptimal approach to analysis of these data. See also the final paragraph of the next item.

5. Data not shared.

Per PLOS ONE, data must either be submitted as part of the paper (as supplementary information), which is not practical for large imaging data such as used here, or the data must be placed in a public repository (such as NITRC). Stating that data are "available from" the senior author upon request (per page 6 of the PDF provided for review) is not adequate.

This reviewer endorses the PLOS ONE data sharing requirement, and asks authors to respect it—in general, for the sake of open science, which accelerates scientific progress.

Furthermore, in the present case, given point 4 above, it is to be anticipated that other researchers may seek to re-analyze the data, using, e.g., group spatial independent component analysis or multilayer community detection, to characterize between-group differences in cortical networks. Rather than assuming that cortical representations are fixed—in spite of an ample literature speaking to “massive functional reorganization in somatosensory cortex” of individuals with amputation—and comparing correlations between specified (atlas-based) spherical regions of interest, as the authors have done here, other researchers may be interested in assessing network spatial distribution and power (using spatial independent component analysis, or other algorithms) and/or network graph theoretic characteristics, such as segregation and recruitment coefficients (using multilayer community detection, or other algorithms). This makes data-sharing even more important.

OTHER CONCERNS:

6. Why were “[c]onnectivity differences … assessed controlling for the influence of age?” (pp. 16 & 18) What is the rationale for this? Were the study data also analyzed without controlling for age?

7. There is some concern about the procedure of flipping images of participants with right-side amputation (“images of participants with a right-side amputation underwent hemisphere inversion across the sagittal midline,” p. 14), since none of the pairs of regions of interest that were used are mirror-symmetric with respect to the midline (per Table 1). This suggests that such flipping could bias outcomes.

8. The slice-timing correction appears to have been performed late in the pipeline (p. 14), which raises concerns. Once images have been coregistered, realigned, and unwarped, performing slice timing correction is problematic, since the data no longer respect the acquisition geometry.

9. Re: “The use of NBS, relying on permutation tests, demonstrated reliability and robustness as an alternative…” (p. 19). This claim raises two concerns: A. No results presented in the present manuscript support the assertion that NBS “demonstrated reliability and robustness” in this study. B. It is not clear what “alternative” means here—what is NBS an alternative to? See also point 3 above.

10. The supplementary table indicates that all participants in both groups were right-handed; this should be stated plainly in the manuscript.

11. (minor) Re: “the Pearson correlation test” (p. 15). The Pearson correlation coefficient is not a “test.”

12. (minor) Re: “ROI-ROI connectivity maps were calculated for each participant…” (p. 15). These were matrices, not “maps.”

Reviewer #2: The study investigates functional connectivity alterations in the sensorimotor network of 26 lower limb amputees using resting-state fMRI. Results show decreased connectivity between the contralateral cerebellum and primary sensorimotor areas, suggesting neuroadaptive changes post-amputation. However, the reviewer has the following major concerns regarding the study’s methodology, data interpretation, and statistical analysis:

1. The manuscript does not provide a power analysis or justification for the sample size of 26 amputees. Given the complexity of functional connectivity analyses, this small sample may lack statistical power.

2. The study does not adequately control for potential confounders, such as years of education, prosthesis use, and amputation duration. These factors could significantly impact functional connectivity patterns.

3. The selection of 10 ROIs lacks a clear rationale. Why were these specific coordinates chosen, and how do they compare to previous studies? More details are needed on how these ROIs represent the sensorimotor network.

4. The analysis reports significant findings at p < 0.05 uncorrected, which is problematic given the number of comparisons performed. A stricter correction, such as Bonferroni or permutation testing, should be applied.

5. The manuscript attributes functional connectivity reductions to neuroadaptive processes without considering alternative explanations, such as motion artifacts or scanner-related issues. Additional validation is required.

6. A cross-sectional design limits interpretations. The manuscript should acknowledge that without pre- and post-amputation comparisons, it cannot definitively attribute changes to amputation itself.

7. Some statistical tests, such as those used for connectivity differences, are described in insufficient detail. The authors should clarify whether parametric or non-parametric methods were used consistently.

8. While the findings suggest connectivity changes, the manuscript does not link these to functional or behavioral outcomes. Claims about clinical significance should be supported by direct evidence.

9. Head motion is a major confounder in resting-state fMRI. The manuscript does not report framewise displacement values or whether high-motion participants were excluded.

10. The manuscript states that 5mm ROIs were used but does not justify this choice. Additionally, a 6mm smoothing kernel may dilute regional specificity, affecting results.

11. The manuscript reports hemisphere inversion for right-sided amputees but does not validate whether this step preserves functional connectivity patterns.

12. Some references (e.g., Koziol et al., 2014 on cognition and the cerebellum) do not directly support the claims being made and should be replaced with more relevant citations.

13. The TR and TE values are given, but essential details such as the field of view, voxel size, and number of volumes per run are not clearly reported.

14. The manuscript presents connectivity findings in Figures 2 and 3 but does not provide full statistical details in figure legends (e.g., exact p-values for non-significant findings).

15. The introduction states a broad hypothesis that connectivity should decrease in amputees, but it does not specify which regions or connections are expected to be affected.

**Do you want your identity to be public for this peer review?** For information about this choice, including consent withdrawal, please see our Privacy Policy

Reviewer #1: No

Reviewer #2: No

---

## [Author Response · Author response to Decision Letter 1]

18 Sep 2025

Overall Assessment

"Cerebellar Modifications of Brain Functional Connectivity in Individuals with Lower Limb Amputation" by Arreguín et al. reports an investigation of differences in functional connectivity between a group of 26 individuals with lower limb amputation, compared to 26 matched healthy control participants, using resting-state functional magnetic resonance imaging. The reported finding of a significant anticorrelation between a spherical region of interest designated as the ipsilateral primary motor cortex and a spherical region of interest designated as the contralateral cerebellar lobule VI may be of some interest.

However, it is not possible to recommend the manuscript for publication as it is currently presented, for the following reasons:

1. The stated conclusions are not supported by the presented results, as most of the reported findings do not reach statistical significance.

2. The finding of a significant anticorrelation does not appear to be consistent with the definition of functional connectivity and therefore requires proper interpretation.

3. The use of the "network-based statistics" (NBS) methodology to calculate only correlations between pairs of regions, rather than broader networks, seems suboptimal.

4. The overall approach—the analytical methodology used and the interpretation of the results—ignores a large body of literature indicating that amputation generally results in "massive functional reorganization in the somatosensory cortex" (see below).

5. The submission does not meet PLOS ONE's data sharing requirement.

General Response

We have carefully considered the observations made on the submitted manuscript and have decided to revise and modify the preprocessing of our analysis. Some of the comments referred to this specific step of the image analysis, and addressing them has led to more precise results regarding the connectivity differences between groups.

Below, we provide a point-by-point response to each comment, detailing the modifications made in the manuscript.

The following modifications were implemented: Initially, during image preprocessing, we performed slice timing correction (STC) after realignment and susceptibility distortion correction using field maps. The modification consisted of performing STC before realignment to prevent any geometric modifications to the images prior to this step (Parker & Razlighi, 2019).

In the original preprocessing, a 6 mm kernel was set for smoothing, considering a 4 mm voxel acquisition. However, we had omitted resampling the images to a 2 mm voxel resolution during normalization to MNI space. We have corrected this by using a 4 mm kernel, following the general recommendation to set a smoothing kernel between 1.5 and 2 times the voxel size (Candemir, 2023; Poldrack, Mumford, & Nichols, 2011).

In the original analysis, subject age was used as a covariate in the calculation of differences. However, we have determined that this is redundant given that the subjects were already matched by age to form the groups. Therefore, the analysis was re-run without this covariate.

As a result of these modifications to the preprocessing and difference analysis, we found similar results. We retained the identification of a significant decrease in the connection between the primary motor region ipsilateral to the amputation and cerebellar lobule 6 contralateral to the amputation (M1.i-CerVI.c). Furthermore, we found a new connection with a significant decrease between the primary sensory region ipsilateral to the amputation and the same cerebellar lobule (S1.i-CerVI.c). Both connections converge on the CerVI.c node, which allows these connections to be considered a subnet of significant connectivity, resulting in a significant decrease in connectivity in the amputation group compared to the non-amputation group. This allows us to discuss changes at the network level and not just a single connection between two nodes within the sensorimotor network.

Finally, the corresponding figures, tables, and text were modified to address the provided observations and analysis changes.

Major Concerns

1.Most results fail to reach statistical significance.

Table 3 lists the "differences in functional connectivity between regions of interest (ROI) between [groups]", but only the first item survives correction for multiple comparisons and is, therefore, statistically significant. The others are not.

Consequently, the claims in the abstract that "statistical analysis revealed a decrease in connectivity in the connections…" (note: "connections," plural) and that "the findings… [highlight] a significant decrease in functional connectivity between the contralateral cerebellum and primary sensorimotor areas…" (note: "areas," plural) are not supported by the presented results, because only one interregional connection was found to differ significantly between the groups.

Response

We appreciate this observation and agree that the original manuscript included imprecise claims regarding our findings by using plural nouns (e.g., "connections," "findings"). After updating the image analysis, we identified a significant network component consisting of two connections: one between the ipsilateral primary somatosensory cortex (S1.i) and the contralateral cerebellar lobule VI (CerVI.c), and another between the ipsilateral primary motor cortex (M1.i) and the same cerebellar lobule (CerVI.c). Both of these connections survived false discovery rate correction (p-FDR < 0.05).

Consequently, we have revised the manuscript to more accurately reflect this finding, using the term "network component" instead of "connections" to avoid unsupported generalizations. Table 3 has also been updated to reflect these findings.

1.1.- The use of ten regions of interest produces a total of 45 possible interregional connections. Therefore, using a p-level threshold of 0.05, without correction for multiple comparisons, it would be expected that a pure noise analysis would generate an average of 0.05 x 45 = 2.25 apparent but not truly significant differences between the groups. Of course, repeating this thought experiment with a new set of pure noise would produce a similar number of apparent but not truly significant differences between the groups, but it would be overwhelmingly likely to be a different set of interregional correlations; such results would not be expected to be replicated.

Response

We appreciate this observation. Both the original and corrected analyses used a connection-level threshold of a T-score > 3 and a False Discovery Rate (FDR) correction (p-FDR < 0.05). This threshold allows us to detect subtle effects while controlling the probability of including false positives in the final connection-level results, which is appropriate for our study. While the arbitrary choice of this free parameter could affect the analysis's sensitivity, the Network-Based Statistics (NBS) method empirically controls this by performing permutations (1000 simulations by default in CONN). We applied FDR correction (p-FDR < 0.05) for between-group comparisons at the connection level and Family-Wise Error (FWE) correction (p-FWE < 0.05) for between-group comparisons at the network level. The reported results survived both corrections, leading us to infer that they are not attributable to chance or false positives. These significance thresholds have now been added to the revised Results Table 3 (p. 8).

1.2.- Table 3 in the current manuscript lists one correlation that is statistically significant (because it survives correction for multiple comparisons), plus five more that are not statistically significant. That the authors report five such insignificant correlations, when the expected average number is 2.25, is of little interest. These same findings are graphically represented in Figures 2 and 3; again, most of what is shown here is not statistically significant, and such insignificant findings cannot adequately support any scientific conclusion.

Response

We appreciate the reviewer's observation and agree that the original version included non-significant data that could lead to misinterpretations. In the corrected manuscript, we've removed the non-significant results from Table 3 to present only those that survived the corrections at both the network and connection levels (p-FDR < 0.05 and p-FWE < 0.05).

1.3 Finally, the manuscript title is not supported by the presented results. Since only one "connection" is significantly different between the amputee group and the reference group, the title's reference to "modifications" (plural) is not appropriate.

Response

After the corrections made to the analysis, we found a difference in a network component composed of two connections between three nodes, which converge on the node identified as CerVI.c. In response to the reviewer's recommendation to modify the title, we propose the following to more accurately reflect the findings in the network component: "Cerebellar Functional Connectivity Alteration in Individuals with Lower Limb Amputation."

2.- Anticorrelation is not connectivity.

The statistically significant finding is an anticorrelation, but the negative sign (the finding of a negative correlation coefficient) is neither mentioned nor discussed. Functional connectivity is defined as synchronous neuronal activity at spatially remote locations, which corresponds to correlation. However, the reported finding is an anticorrelation (a correlation coefficient with a negative sign). This does not align with the definition of functional connectivity. Instead, it resembles the well-known temporal anticorrelation of activity in the default mode network with respect to activity in task-positive regions. The finding is of some interest, but its sign is important: the anticorrelation must be presented and discussed as such, not ignored and treated as if it were a positive correlation.

Response

We appreciate the reviewer's observation. In the corrected version of the manuscript, we have used the term "anticorrelation" (e.g., "This subnetwork was specifically composed of two anticorrelated connections…," p. 11) when referring to the significant connection found between the evaluated regions.

Additionally, we have included a specific discussion on the meaning of negative correlations in resting-state fMRI studies. We have noted that while functional connectivity has been traditionally associated with positive correlations (Friston, 1993), anticorrelations can also reflect inhibitory neuronal processes or complementary inverse interactions in the hemodynamic response between regions (Goelman et al., 2014).

3. Network-based statistics is questionable when restricted to pairs of regions.

Network-based statistics is analogous to the more familiar and well-established cluster-based statistics. The idea of cluster-based statistics is that instead of testing the null hypothesis at every voxel, we test the null hypothesis for each cluster of potentially active voxels. The advantage is increased power; the disadvantage is that we don't know exactly where in the cluster the activity is; we know that there is activity in the cluster, but not precisely where in the cluster.

Network-based statistics is similar: instead of testing the null hypothesis at every voxel, we test the null hypothesis for each network of voxels (or, more generally, nodes). This yields more power, but at the expense of not knowing exactly where within a "hot" network the "activity" truly resides. We can say that the network is hot (or significant), but not where in the network the activity or significance resides.

What is key in both cases is the use of clusters or networks of substantial size. It would be odd to use cluster-based statistics but to limit the analysis to clusters composed of only two voxels. Yet, unless this reviewer is mistaken, the present manuscript appears to do this with network-based statistics. It appears that only correlations between two regions are considered; these are networks composed of only two nodes. This seems a suboptimal use of network-based statistics.

Response

We appreciate the reviewer's observation regarding the use of the NBS method. We would like to clarify that our analysis was not limited to pairs of regions but was performed on the whole functional connectivity matrix of 10 × 10 regions of interest (ROIs), which results in a total of 45 possible connections.

The NBS method (Zalesky et al., 2010) was applied to this entire network to identify subnetworks of connections that, collectively, showed a significant difference between the two groups. The result of this analysis was a significant network component (a subnetwork) composed of three nodes and two connections that survived the strict statistical corrections for type I error at the network level (p-FWE < 0.05) and the connection level (p-FDR < 0.05). This finding is not an artifact of a restricted analysis but the result of the rigorous method we employed.

We believe that the use of NBS was the most appropriate and robust methodological choice for our study for several reasons:

● Non-parametric analysis: The NBS method does not assume a normal distribution of the data, which is ideal and safer for our dataset, as it is not large enough to meet parametric assumptions reliably.

● Control of false positives: NBS controls the family-wise error rate (FWE) across the entire network, thereby drastically reducing the risk of obtaining false positives compared to analyzing individual connections.

● Network-level analysis: Although the resulting subnetwork was small, the fact that the network-level analysis identified it as the only significant component gives it a biological validity and meaning that could not be achieved with a pair-wise correlation analysis.

We hope this clarification is helpful and gives you confidence in the robustness of our methodology. In response to the observation, we modified the method description in the manuscript for a better understanding of the method: "The Network-Based Statistic (NBS) is a non-parametric method designed to identify differences..." (p. 8).

4."Reassignment" has been ignored.

It is well established that "peripheral denervations of large expanses of the sensory epithelium, as a result of limb amputation… can lead to massive functional reorganization in the somatosensory cortex…" (from "Thalamic and Cortical Contributions to Neural Plasticity after Limb Amputation" (Florence et al., 2000)).

Indeed, the introductory section of the present manuscript (p. 10) refers to "…cortical functional reorganization after amputation…"

However, the analysis in the present manuscript ignores such functional reorganization, as it is limited to calculating correlations between spatially smoothed MR signals averaged in ten-mm spheres located at fixed positions in neuroanatomical atlas space. The "massive functional reorganization in the somatosensory cortex" of individuals with the amputation is simply ignored. This approach is not optimal for the analysis of these data. See also the last paragraph of the next point.

Response

We appreciate the reviewer's comment and agree that post-amputation cortical reorganization is a widely documented phenomenon, particularly in the somatosensory cortex contralateral to the amputated limb (Flor et al., 2006; Makin et al., 2013). However, the objective of this study was not to map this reorganization but to evaluate possible alterations in functional connectivity between cortical representations of lower limbs previously identified in other studies on people without amputations. To this end, we selected ROIs in MNI space located in the bilateral motor and somatosensory cortex. We acknowledge that this design does not allow for a direct evaluation of cortical remapping, which has been indicated as a perspective for future studies within the discussion: "It is important to acknowledge that the present analysis did not incorporate potential cortical reorganization prior to the evaluation of functional connectivity, a factor that should be considered in future projects" (p. 15).

5.Data Not Shared.

According to PLOS ONE, data must either be submitted as part of the article (as supplementary information), which is impractical for large image datas

---

## [Decision Letter · Decision Letter 1]

26 Nov 2025

Cerebellar functional connectivity alteration in individuals with lower limb amputation

PONE-D-24-44413R1

Dear Dr. Paredes,

We’re pleased to inform you that your manuscript has been judged scientifically suitable for publication and will be formally accepted for publication once it meets all outstanding technical requirements.

Kind regards,

Usman Ghafoor

Academic Editor

PLOS ONE

Additional Editor Comments (optional):

The authors have adequately improved the manuscript that warrants its publication.

Reviewers' comments:

Reviewer's Responses to Questions

**Comments to the Author**

Reviewer #1: (No Response)

Reviewer #3: All comments have been addressed

2. Is the manuscript technically sound, and do the data support the conclusions?

Reviewer #1: Partly

Reviewer #3: Partly

3. Has the statistical analysis been performed appropriately and rigorously?

Reviewer #1: I Don't Know

Reviewer #3: Yes

4. Have the authors made all data underlying the findings in their manuscript fully available?

Reviewer #1: No

Reviewer #3: Yes

5. Is the manuscript presented in an intelligible fashion and written in standard English?

Reviewer #1: Yes

Reviewer #3: Yes

Reviewer #1: Please make the data public, so that other investigators can use it to investigate functional remapping.

Reviewer #3: The authors have thoroughly addressed and resolved all the concerns raised in the previous round of review. They have provided detailed clarifications, strengthened the methodological explanations, and incorporated the suggested revisions throughout the manuscript. Based on the improvements made and the enhanced clarity and rigor now reflected in the revised manuscript, the reviewer is satisfied that the work meets the necessary standards for scholarly publication. Therefore, the reviewer endorses the manuscript for acceptance and recommends proceeding with publication.

**Do you want your identity to be public for this peer review?** For information about this choice, including consent withdrawal, please see our Privacy Policy

Reviewer #1: No

Reviewer #3: No

---

## [Editor Report · Acceptance letter]

PONE-D-24-44413R1

PLOS ONE

Dear Dr. Paredes,

I'm pleased to inform you that your manuscript has been deemed suitable for publication in PLOS ONE. Congratulations! Your manuscript is now being handed over to our production team.

Kind regards,

on behalf of

Dr. Usman Ghafoor

Academic Editor

PLOS ONE